# Optimization and Validation of Efficient Models for Predicting Polythiophene Self-Assembly

**DOI:** 10.3390/polym10121305

**Published:** 2018-11-26

**Authors:** Evan D. Miller, Matthew L. Jones, Michael M. Henry, Paul Chery, Kyle Miller, Eric Jankowski

**Affiliations:** 1Micron School of Materials Science and Engineering, Boise State University, Boise, ID 83705, USA; evanmiller326@boisestate.edu (E.D.M.); mattyjones@boisestate.edu (M.L.J.); mikehenry@boisestate.edu (M.M.H.); 2Physics, Macalester College, St. Paul, MN 55105, USA; paulachery@gmail.com; 3Physics, University of Puget Sound, Tacoma, WA 98416, USA; kyledanielmiller@gmail.com

**Keywords:** molecular dynamics, organic photovoltaics, coarse-graining

## Abstract

We develop an optimized force-field for poly(3-hexylthiophene) (P3HT) and demonstrate its utility for predicting thermodynamic self-assembly. In particular, we consider short oligomer chains, model electrostatics and solvent implicitly, and coarsely model solvent evaporation. We quantify the performance of our model to determine what the optimal system sizes are for exploring self-assembly at combinations of state variables. We perform molecular dynamics simulations to predict the self-assembly of P3HT at ∼350 combinations of temperature and solvent quality. Our structural calculations predict that the highest degrees of order are obtained with good solvents just below the melting temperature. We find our model produces the most accurate structural predictions to date, as measured by agreement with grazing incident X-ray scattering experiments.

## 1. Introduction

Limiting the negative impact of global climate change requires a shift to sustainable energy generation. Organic photovoltaics (OPV) are carbon-based solar panels that could meet this need with one-day energy payback times, if they can be manufactured at scale with sufficient (∼15%) power conversion efficiency (PCE) [1]. The low cost of OPVs derives from the solution processability of their active ingredients, which can also be used to make light emitting diodes and field effect transistors [2,3]. Controlling the solution-phase self-assembled morphology—the microstructure that emerges spontaneously during fabrication—is central to meeting the 15% efficiency target, as morphology governs performance [4]. The choice of organic electronic compounds, solvents [5], annealing protocols, [6,7,8] and processing temperatures [9] all affect the self-assembled microstructure and resultant device characteristics. However, the impracticality of exhaustively testing which chemistries and processing protocols are sufficient to manufacture environment-friendly devices [10] necessitates computational or theory-based guidance of promising candidates.

Poly-3(hexylthiophene) (P3HT) is one of the polymers that kick-started research into solution-phase self-assembly of OPVs [11,12], and is the focus of this work. Often referred to as the “bench-mark” OPV compound [13], the relative ease of working with P3HT has led to hundreds of studies linking P3HT’s structure to its performance in OPV devices [14]. This work on P3HT over nearly two decades highlights the difficulty and opportunity in optimizing self-assembly, and underscores the potential utility of informing experimentation with theory. How many of these experiments could have provided additional insight if equilibrium phase diagrams of P3HT were known in 2002? Would more promising ingredients have been identified earlier if a theoretical maximum PCE for P3HT blends was known? Answering these questions requires calculating phase diagrams and predicting PCE, which requires models of P3HT capable of predicting self-assembly.

Predicting P3HT self-assembly in particular, and OPV assemblies in general, is difficult because of the multiple length-scales that matter: atomic orbitals, molecular packing, alignment of crystallites, and thermodynamic phase separation all impact OPV device performance. First principles calculations have the highest resolution and can provide insight into charge transport relationships in P3HT [15,16,17], but their computational demands preclude simulating thousands of atoms—far too small to gain insight into the bulk morphological features that arise from thermodynamic self-assembly. Macroscopic models are successful in predicting device-scale morphologies with thickness ∼100 nm both on-lattice [18,19,20,21,22] and off-lattice [23,24,25], but cannot represent important structural features such as crystallite grain orientations and energetic differences between molecules. Molecular models implemented in molecular dynamics (MD) or Monte Carlo simulations fill the gap between first-principle and macroscopic models, though the system size versus relaxation time trade-off significantly hinders investigations of self-assembly [26,27,28,29,30,31,32,33,34]. At the largest scales, the structural evolution of 5 million coarsely-modeled P3HT monomers can be accessed on >100 nm length-scales [34], but the computational cost of evaluating each step meant that equilibration was inaccessible over the 400 ns simulation trajectory. At 11-nm scales, equilibration of coarse-grained P3HT models is achievable over ∼2 μs simulation trajectories [28], but such coarse models miss the π-stacking details of P3HT rings, which can have implications for charge transport calculations [35,36]. Long relaxation times can be avoided in MD simulations through carefully selected initial conditions [32,37,38,39,40], but these simulations can only check if a structure is locally stable, not whether it will robustly self-assemble at a particular state-point. Determining the optimal “sweet spot” between system size, model resolution, and computational cost of predicting equilibrium is therefore essential if MD simulations of thousands of candidate materials and conditions are to be used to inform OPV experimentation [41].

Validating predictions of OPV structures against experiments further complicates the challenge of building good simulations because the two most common characterization techniques—surface analysis and reciprocal space spectroscopy—each provide an incomplete picture of structure [42,43]. Surface techniques such as atomic force microscopy detect crystalline and amorphous regions at the surface of the film but do not reveal structural characteristics important for charge generation and transport within the film [44]. Reciprocal space techniques such as grazing incident X-ray scattering (GIXS) probe the bulk material, revealing averaged periodic features but lack the precision to resolve a unique solution of molecular positions. Additionally, analysis is complicated by residual solvents and amorphous regions [42]. In concert, surface and scattering experiments provide references against which simulations can be checked, but doing so requires the modeling of volumes large enough for features that repeat on 10 Å to 30 Å scales, reinforcing the need for simulations that are big enough and fast enough.

The aims of this work are to (1) describe an optimized model of P3HT that is efficient and meets structural prediction needs, (2) resolve ambiguity around what “big enough” and “fast enough” means for P3HT, and (3) discuss transferable recommendations for simulating other OPV materials. This work is organized as follows: We present our P3HT model in Section 2 and characterization techniques in Section 3. We explain the important performance metrics and discuss current and future requirements for predicting OPV self-assembly in Section 4.1. We employ small-scale simulations to evaluate P3HT self-assembly over ∼350 state-points in Section 4.2 and evaluate the impact of simulated solvent evaporation in Section 3.1. We evaluate structural predictions with large-scale simulations in Section 4.4, and finally validate against experimental measurements in Section 4.5.

## 2. Model

We represent P3HT molecules with a “united-atom” model [45] wherein hydrogen atoms are treated implicitly through the interactions between three types of simulation elements (“beads”): aromatic carbon (CA), aliphatic carbon (CT), and sulfur (S) (Figure 1). This level of coarse-graining is convenient for modeling OPV materials, as the reduction of simulation elements from 25 atoms to 11 united-atom sites per monomer reduces computational cost [45,46,47], while simplifying back-mapping of atomic coordinates for charge transport calculations [35]. United-atom models have been used to successfully predict structures for a variety of systems including: polymers [48,49], proteins [46], and small molecules [50,51]. The base units of mass M=32 amu, energy E=0.32 kcal/mol, and length L=3.905 Å, used to describe interactions within the simulation, are adapted from the Optimized Potentials for Liquid Simulations (OPLS)-United Atom (UA) force-field [52]. The pairwise non-bonded interaction potentials derived from these base units are presented in Table 1 in which ϵ is the depth of the Lennard–Jones potential, $\sigma_{LJ}$ is the van der Waals radius, and *m* is the mass of the bead.

The pairwise bonded constraints (bond lengths, triplet angles, and quadruplet dihedrals) are taken from a modified atomistic force-field based on OPLS-All Atom, as parameterized by Bhatta et al. [37]. Since this force-field is atomistic, we adapt it to account for implicit hydrogens and the reduced number of element types in our model (see Appendix A for full details). We also model aromatic thiophene rings as rigid: the bonds, angles, and dihedrals are fixed, maintaining the relative positions of the elements of the rigid bodies throughout the simulation [53]. Abstracting away these degrees of freedom reduces the number of calculations required for each timestep while maintaining good agreement with experiment for both the chemical and structural properties of a variety of molecule types [49,50,54,55,56]. We further optimize this model by adjusting interaction parameters to better predict P3HT structure, and lower computational cost with implicit solvent and electrostatics.

The unit of time, *t*, can be calculated from the base units:(1)$$t=\sqrt{\frac{ML^{2}}{E}}=1.8\times 10^{-12}\;s.$$

We use a timestep of 0.001t, so each simulation timestep corresponds to 1.8 fs. The “base case” model considered here handles solvent and long-range electrostatics implicitly, and each oligomer comprises 15 monomers (15mers). Comprehensive evaluation of the optimized model assumptions, including explicit consideration of electrostatic interactions and short oligomer chains are included in Appendix A.

## 3. Methods

In this work, we conduct molecular dynamic (MD) simulations using the Graphical Processing Unit (GPU)-accelerated HOOMD-blue simulation package [57,58], performed on NVIDIA K80 and P100 GPUs. The code used to produce this data is open-source and freely available at Ref. numbers [59]. The complete dataset from this investigation is available at Ref. numbers [60]. Simulations are conducted in the canonical ensemble (NVT), in which the total number of particles, volume, and temperature are kept constant. The Nosé–Hoover thermostat, in which the system is coupled to a heat bath, is applied to maintain the temperature [61]. Particle positions and velocities are updated with the two-step velocity-Verlet integration of Newton’s equations of motion with a timestep of 1.8 fs [62].

Each simulation is initialized from a unique random configuration within a cubic volume with periodic boundary conditions. We accomplish this by first placing molecules created with the mBuild software package [63] at random positions in a large simulation volume, where molecules are sufficiently separated so that they can be placed without overlapping their neighbors. A short MD simulation (1.8 ns) is performed at high temperature (T ∼ 1300 K) to randomize the molecule positions and orientations. The system volume is then reduced during another short simulation (1.8 ns and 1300 K) until the target density is reached. This process of “initializing”, “mixing”, and “shrinking” has been previously used to initialize independent snapshots at arbitrary densities [28,49,50,64]. Unless otherwise specified, every simulation presented herein is instantaneously quenched from high temperature to the target temperature for the duration of its NVT simulation. We consider target temperatures from T=80 to 1300 in steps of 80 K. These temperatures span the glass transition (300 K) and melting (490 K) temperatures expected for P3HT [37]. Of course, real P3HT degrades at the higher temperatures in this range. Our simulations at these high-*T* conditions are performed to provide unique independent snapshots from which to initialize independent simulations, and to check if high-temperature structural transitions might exist if P3HT did not degrade.

We consider a range of relevant film densities, ρ=0.56,0.72,0.89,1.05,1.11 g/cm^3^, with the largest ρ=1.11 g/cm^3^ corresponding to the experimental thin-film density for P3HT [65]. For the lowest densities, as much as 40% of the simulation volume is occupied by the implicit solvent, whereas the volume occupied by the implicit solvent is negligible for the highest densities. We employ an extremely simplified model of solvent quality: we define the parameter $\varepsilon_{s}$ to represent how poor the solvent is for P3HT, and scale all of the pairwise interaction potentials by this amount ($\epsilon_{ij}\to \varepsilon_{s}\times \epsilon_{ij}$). We explore $\varepsilon_{s}$ values from 0.2 to 1.2 in this work. Low values of $\varepsilon_{s}\leq 0.7$ correspond to solvents in which P3HT is highly soluble (e.g., chloroform, chlorobenzene or 1,2-dichlorobenzene), whereas larger values $\varepsilon_{s}>0.7$ describe solvents where P3HT is less soluble (e.g., acetone) [8]. While this simplified model cannot capture complex or entropic solvent phenomena, it provides a significant computational advantage [66,67,68]. Furthermore, adjusting $\varepsilon_{s}$ while holding *T* constant enables exploration of how equilibrium structure depends on molecular attractions at fixed kinetic energy. In this work, we perform simulations at the combinations of *T*, ρ, and $\varepsilon_{s}$ described above to understand how these parameters in concert influence thermodynamic phase behavior.

### 3.1. Solvent Evaporation

Each simulation performed herein utilizes one of two simulation protocols to sample microstates at the target state-point. Protocol (1) ignores solvent evaporation: disordered initial configurations at the target density, ρ, and solvent quality, $\varepsilon_{s}$, are instantaneously quenched from T=1300 K to the target temperature, after which equilibration progress is monitored [50]. Protocol (2) is a very basic, qualitative model of solvent evaporation that helps to sample configurations at experimental densities (ρ=1.11 g/cm^3^): First, a system is equilibrated at ρ=0.72 g/cm^3^ and the target temperature and $\varepsilon_{s}$ using Protocol (1), followed by a linear compression to ρ=1.11 g/cm^3^ over 280 ns. After the shrinking step of Protocol (2), equilibration progress is monitored as in Protocol (1).

Modeling implicit solvent removal in this way creates two fundamental tensions with our claimed thermodynamic approach. Firstly, invoking Protocol (2) suggests that microstate sampling with Protocol (1) is non-ergodic over practical time scales. It is well established in both experiments and simulations that polymer dynamics are kinetically arrested at higher densities, so Protocol (2) can be viewed as a sampling acceleration scheme that assumes structures arising from enthalpy minimization at low densities are representative of equilibrium at high densities. Of course, steric entropic effects are known to contribute most interestingly to the free energy at high densities, with the striking diversity of entropically stabilized hard-polyhedra phases as just one example [69]. This leads to the second tension: Invoking an implicit solvent model assumes entropic contributions of the solvent either (a) can be effectively represented in the coarse Hamiltonian, or (b) are negligible compared to enthalpic contributions. It is therefore implied in the present work—and every multiscale study invoking coarse-graining—that potential energy minimization dominates the free energy minimization of the coarse Hamiltonian, whose emergent coarse structures represent the underlying atomistic description with all of its encoded entropic contributions. In the cases where such coarse-graining is not predictive of the more detailed representation’s structure, there are interesting open questions about how to include entropic contributions within the coarse Hamiltonian explicitly, or whether back-mapping fine-grained structure is sufficient, both beyond the scope of this work.

Protocol (2) does not capture evaporation-driven dynamics that occur in real systems, including the alignment of linear molecules induced by hydrodynamic flows and steric effects at interfaces. In principle, such dynamics could be used to facilitate self-assembly, but are beyond the equilibrium approach of this work.

### 3.2. Morphology Characterization

To characterize the molecular packings obtained in our simulations we use two structural metrics: an order parameter and simulated GIXS using the Diffractometer simulation software [35,70]. GIXS patterns are used to identify and quantify periodic morphological features and are used to validate predicted structures directly against experiments. We obtain a set of patterns by simulating diffraction on each cubic morphology from 60 unique orientations uniformly distributed on a sphere. We identify orientations with clearly resolved peaks and align crystallographic directions along the same axes before averaging these orientations into a final diffraction pattern. Treating the diffraction patterns in this way improves signal-to-noise ratio of periodic features, allowing detection of periodic length-scales more precisely.

The order parameter, ψ, is used to describe the proportion of thiophene rings in “large” clusters. The clustering algorithm is described in full in our previous work and presented, with examples, in the Appendix A [50]. Briefly, two thiophene rings are considered “clustered” if their centers-of-mass are within 6.6 Å of each other and if the planes of the thiophene rings are oriented within 20° degrees of each other. The value of 6.6 Å is informed by the radial distribution function of the thiophene centroids in ordered P3HT, and the 20° cut-off is taken due to rotations under this having small effect on the transfer integral (a measure of the electron orbital overlap) between two rings [16]. A cluster must contain at least six thiophene rings to be considered “large” and contribute towards ψ, a cut-off that is selected to distinguish morphologies with fewer large clusters from those with many small clusters.

## 4. Results and Discussion

Here, we benchmark P3HT simulations using our optimized model to provide context for the system sizes that are practically accessible, perform experiments with simulated solvent evaporation as potential way to avoid long relaxation times, and evaluate the system sizes needed to validate predictions against experiments.

### 4.1. Computational Performance and Scaling

The time it takes to predict self-assembly of a material with MD primarily depends upon the size of the simulated volume, which affects two key metrics:Relaxation time: The number of timesteps that must be evaluated before the system reaches equilibrium. Larger volumes generally mean larger relaxation times because more molecules must rearrange before the system has converged to the equilibrium distribution of microstates.Computational performance: The number of timesteps that can be evaluated per each second that elapses on a clock on the wall, here measured as Timesteps Per Second (TPS). TPS scales between $O(N^{-1})$ and $O(N^{-2})$.

We measure relaxation time and TPS in order to quantify the practicality of performing equilibrium simulations as a function of system size. We perform instantaneous quenches to T=600 K at ρ=0.72 g/cm^3^ and $\varepsilon_{s}=$ 0.8 for our base case model with *N* ranging from *N* = 16,500 to *N* = 600,000.

Figure 2a shows TPS decreases monotonically with *N*, closely matching the $O(N^{-1})$ reference slope (orange). For the smallest systems (*N* = 16,500), this corresponds to being able to perform 400 ns per day, and, for the largest systems (*N* = 600,000), 17 ns per day. In Figure 2b, we show a characteristic time evolution of the Lennard–Jones pair potential energy, which we use as one proxy for structure. At equilibrium, measurements of potential energy are observed to fluctuate about a stable, time-invariant average (Region 3 in Figure 2b). Before equilibrium is reached, we observe a fast initial change in structure (Region 1), followed by a slower relaxation time (Region 2). We detail the automatic detection of these regions and present the curves for multiple systems in Appendix A. Here, we observe Region 1 is insensitive to *N*, occurring within the first 0.5 μs of simulation time. The relaxation time (Region 2), however, strongly depends on *N*. We measure relaxation times of ∼0.2 μs for the ∼16,000 beads, ∼0.4 μs for ∼29,000 beads, and ∼1.0 μs for 40,000 beads. For system sizes larger than *N* = 40,000, we do not observe equilibration of the base case model 15mers at T=600 K, ρ=0.72 g/cm^3^, $\varepsilon_{s}=0.8$. Empirically, these observations suggest relaxation time scales $O(N^{2})$, though the longer relaxation times for larger *N* tested here preclude detailed evaluation.

Once a system has come to equilibrium, we measure decorrelation times, explained in detail in previous works [49,50]. The 100 15mer simulation requires ∼80 ns for each independent measurement to be generated, the 175 15mer simulation requires ∼50 ns per measurement, and the 250 15mer simulation requires ∼300 ns per measurement. Therefore, simulations of around one μs are needed to sample the equilibrium distribution of microstates after the relaxations of Regions 1 and 2.

To put these performance numbers in context with advances in computational hardware, we benchmark P3HT systems with *N* = 165,000 on four different Nvidia GPUs and six different high performance computing systems. Figure 2c shows a factor-of-two improvement in TPS roughly every two years. The TPS scaling and relaxation scaling as a function of *N*, and evolution of TPS over GPU release year data presented so far allows us to answer “How many years must we wait before we can equilibrate a system with twice the spatial dimensions of the largest practical dimensions today?” Doubling the size of the simulation volume along each axis results in eight times the volume and therefore eight times higher *N* (given the same density), so we would expect the TPS to drop by a factor of ∼8, given that TPS $\propto O(N^{-1})$. However, as relaxation time scales as roughly $O(N^{2})$, we would require 64 times as many simulation time steps to equilibrate before sampling. This means that doubling the linear dimensions of a system requires 512 times the TPS to equilibrate it in the same amount of time. Extrapolating current hardware trends, a new GPU 18 years from now would meet this performance need. Of course, performance scaling will vary significantly, depending on model details (e.g., chain length), *T*, ρ, and $\varepsilon_{s}$, so the precise numbers reported here will have limited transferability to other chemistries and conditions.

Even so, we draw two takeaways from these data: The first dispels the idea that significantly larger volumes can be equilibrated with incremental advances in hardware. Rather, doubling the dimensions of a system requires decades of hardware improvement, all other factors being equal. Consequently, the second takeaway is that techniques that mitigate relaxation times will be essential in predicting OPV morphologies relevant to device scales. Such techniques include modeling at multiple scales, modeling the minimal necessary physics at each scale, efficiently sampling parameter space, and advanced sampling techniques [71,72,73].

### 4.2. Identifying Optimal Assembly Conditions

Despite the divergent behavior of simulation time as a function of *N* observed in the previous section, it is computationally tenable to efficiently sample the state space of P3HT self-assembly using a base case system of 100 15mers, using our OPLS-UA model. We therefore perform an ensemble of MD simulations over a range of 350 unique state-points (depicted by a black “x” in Figure 3) each defined by *T*, ρ, and $\varepsilon_{s}$, to determine which combinations are correlated with self-assembly. Doing so generates the rough phase diagram of P3HT structure as a function of *T*, ρ, and $\varepsilon_{s}$. Each simulation employs cubic volumes with edge length ∼7 nm. These volumes relax to equilibrium within ∼180 ns, after which the ordering measured by ψ is constant. Decorrelated equilibrium microstates are drawn from trajectories after this initial relaxation, with an additional 180 ns of simulation time generating ∼8 microstates per state-point. The colorbars in Figure 3 quantify the degree of ordering measured by the order parameter ψ. In each case, more ordered systems appear red, whereas systems with less ordering and fewer ordered clusters appear in blue. The order parameter values depicted between simulated state-points are linearly interpolated.

In Figure 3, we observe two major trends in P3HT ordering: (1) increasing the density limits the ordering and (2) there exists a narrow band of *T*-$\varepsilon_{s}$ combinations that produces a high degree of order, independent of density. The first trend arises from systems becoming kinetically arrested: Chains have little room to rearrange at high densities after being instantaneously quenched below the melting temperature. The second trend arises from the relationships between T, kinetic energy, and the scaling of the Lennard–Jones well-depths through $\varepsilon_{s}$. When the ratio $\frac{T}{\varepsilon_{s}}$ is sufficiently high, simulation elements have sufficient kinetic energy to routinely break out of the short-range pairwise potential energy wells of their neighbors. Conversely, in systems with deep potential wells, beads are more likely to get stuck in local potential energy minima, resulting in longer relaxation times. As expected, we observe that P3HT orders most robustly when it has both sufficient free volume and kinetic energy to rearrange, providing the temperature is below the melting temperature. These requirements are consistent with experimental annealing practices used to increase order, where energy is added (thermal annealing) or interaction strengths are decreased while increasing free space for polymers to rearrange (solvent annealing).

In Figure 3, we also observe that P3HT is able to robustly self-assemble over a range of a couple of hundred Kelvin. This self-assembly occurs just below the melting temperature, given a particular solvent quality. In systems with sufficient free space to order, the model predicts melting temperatures in the range of ∼400 to 600 K (depending on the solvent strength), which corresponds well to the experimentally observed melting temperature at 490 K. The experimental melting temperature of P3HT in the absence of a solvent is reproduced when $\varepsilon_{s}=0.5$, indicating that the optimized OPLS force-field used here overpredicts P3HT’s melting temperature and that varying $\varepsilon_{s}$ can be thought of as either varying solvent quality, or correcting for systematic attraction offsets in the force-field.

### 4.3. Modeling Solvent Evaporation Facilitates Equilibration

We observe that P3HT simulations at ρ≥1.05 g/cm^3^ show a low degree of order, ψ (Figure 3), when instantaneously quenched from T=1300 K to the target simulation temperature. However, highly ordered P3HT has been observed in experiments at and near this density. We explain this discrepancy by kinetic arrest over simulation timescales: Closely-packed P3HT volumes with negligible solvent have long rearrangement times. To avoid such trapping and to more faithfully model solvent evaporation, we perform “shrinking” simulations using a simple model of solvent evaporation (Protocol (2)) from ρ=0.72 g/cm^3^ to ρ=1.11 g/cm^3^ over 36 ns, and compare the resultant systems to the base case in the previous section. The initial density ρ=0.72 g/cm^3^ is chosen because it is the highest density at which highly-ordered morphologies are robustly assembled. When solvent evaporation is modeled in this way we generally observe negligible change in ψ (Figure 4) as the system transitions from ρ=0.72 g/cm^3^ to ρ=1.11 g/cm^3^. At high temperatures, 600≤T<900 K, we observe increased ordering as a result of solvent evaporation (Figure 4), which is consistent with previous work showing that increased density at constant temperature can lead to a higher degree of order [50]. In aggregate, these results indicate that our OPLS-UA model is efficient enough to identify the temperature-solvent-density combinations that result in molecular self-assembly.

The results presented in Figure 3 are generated with Protocol (1): Low-solvent (high P3HT density) systems display less order because of longer rearrangement times. The results presented in Figure 4 are generated with Protocol (2): Equilibrating and then shrinking the simulation volume while holding temperature constant results in structures that are as ordered as those at ρ=0.72 g/cm^3^, but at the experimental density of ρ=1.11 g/cm^3^, and with GIXS in quantitative agreement with experiments (Figure 6). We therefore recommend using Protocol (2) for simulating solvent evaporation where appropriate because, otherwise, long rearrangement times at high densities can be avoided.

### 4.4. Large Volumes Are Needed for Experimental Validation

Here, we combine the results of the previous two sections and perform solvent evaporation simulations of large volumes at specific state-points to evaluate which advantages in structural insight, if any, are afforded with larger volumes. We compare the base case “small” systems of 100 15mers (*N* = 16,500, L=7 nm) against “large” 1000 15mer (*N* = 165,000, L=15 nm) systems. The large simulations are initialized at T=600 K, ρ=0.72 g/cm^3^, and $\varepsilon_{s}=0.8$ using Protocol (1). During the evolution of the large systems, we record atom positions at three different degrees of order: when the system is disordered, when some crystallites have formed but disordered regions still exist (semi-ordered), and when it has ordered. These times are chosen based on the degree of structural evolution discussed in Section 4.1. Each of these snapshots is used to initialize independent simulations using Protocol (2) to reach ρ=1.11 g/cm^3^ over a 180 ns simulation trajectory. We compare these three large morphologies at experimental densities, “disordered”: ψ∼ 0.4, “semi-ordered”: ψ∼ 0.6, and “ordered”: ψ ∼ 0.8), to the smaller base cases. Note that we present our analysis for only the ordered system here in the main text and in Appendix A we present the analysis for the semi-ordered and disordered systems.

The large and small ordered systems shown in Figure 5a,b, in which only “large” clusters, identified using the cluster analysis discussed in Section 3.2, are shown (large clusters ≥ 6 monomers, and side chains are omitted). The large ordered system contains a few large crystallites, colored blue, red, and yellow. This contrasts with the small morphology, which primarily consists of a single large crystallite (shown in blue in Figure 5a), with the next largest having significantly fewer members (shown in red). These results indicate that smaller systems will tend to have fewer ordered crystallites, which limits the opportunity to observe periodic organization of these structures. Despite this difference, GIXS patterns show that the same periodic distances are present in both system sizes (Figure 5c vs. Figure 5d), albeit with significantly increased noise in the case of the smaller system. As such, small morphologies can be used to identify state-points of structural interest; however, large simulations are better at characterizing crystal structure and quantifying morphological order.

### 4.5. Experimental Validation of Optimized P3HT Model

To validate our model, we perform simulations of 1000 15mers with Protocol (2) and compare simulated GIXS patterns against experimental P3HT patterns (T=600 K, $\varepsilon_{s}=0.8$). Predicted and experimental GIXS patterns are presented in Figure 6a (averaged over 18 simulation orientations) and Figure 6b (Reprinted with permission from numbers [74]. Copyright 2012 American Chemical Society). Both experimental and predicted structures are characterized by bright reflections extending vertically along the out-of-plane axis with reciprocal spacing of 0.38 ± 0.02 Å^−1^ (corresponding to real-space separation of 16.5 Å) and the narrow peak perpendicular to the [100] direction at 1.68 ± 0.02 Å^−1^ (corresponding to a real-space separation of 3.74 Å). To connect these scattering features to morphological features, we present the ordered morphology in Figure 7a, which shows lamellae of π-stacked thiophene rings (shown with dark blue CA and yellow S), and aliphatic tails (cyan CT). It is the periodic π-stacking at ∼3.7 Å and perpendicular alkyl-stacking at 16.5 Å responsible for the ∼1.7 and 0.4 Å^−1^ features that are observed in the GIXS patterns. The agreement between experimental and predicted structures demonstrates the present OPLS-UA model is capable of efficiently and quantitatively predicting ordered P3HT structures within three weeks of simulation on a single GPU. Also similar to the structures seen experimentally, the lamellae in the ordered system do not represent a single, perfect crystal, but rather multiple crystallites with various grain orientations. The thiophene rings in these grains are depicted by red, blue, and yellow in Figure 7b.

Within each layer, the thiophene rings primarily stack co-facially in either an “aligned” (Figure 7c) or “anti-aligned” (Figure 7d) conformation, in which the sulfur atoms of adjacent rings are on the same side or opposite sides of the stack respectively. We calculate the radial distribution function (RDF, Figure 7e) between monomer centers to characterize short-range packing. A monomer center is defined by the geometric average position of the sulfur and two furthest carbons in the thiophene rings (see Figure 7e inset), and the spacing between two centers is used to distinguish aligned and anti-aligned π-stacking. The first peak in the RDF describes π-stacking of the thiophene heads and is split into two features at 3.9 and 5.3 Å corresponding to the aligned and anti-aligned cases, respectively. As made evident by the RDF peak magnitudes, we observe a slight preference for aligned thiophene stacking vs. anti-aligned stacking. Generally, more ordered morphologies of P3HT are expected to provide faster charge transport characteristics. As such, these results show that sufficient amounts of good solvent, which is then evaporated off just below the P3HT melting temperature, are expected to produce ordered morphologies with beneficial electronic device properties.

## 5. Conclusions

In this work, we presented insight into semiconducting polymer assembly aimed at both molecular simulators and experimentalists. Specifically, for P3HT, we demonstrated excellent quantitative agreement with P3HT nanostructure investigated by GIXS and we found temperature and solvent combinations where robust self-assembly into ordered structures is expected. In doing so, we validated the predictive accuracy of our optimized OPLS-UA model, which implicitly includes solvent, charges, and abstracts away fast degrees of freedom in the thiophene rings.

Our analysis of computational efficiency scaling with simulation size showed that projected improvements to computational hardware over the next decades will not enable the equilibration of significantly larger organic semiconductor volumes than those presented, using the techniques demonstrated in this work. Since relaxation times were identified as the limiting factor to polymer equilibration, multi-scale techniques and model approximations must be used in order to predict OPV morphologies at experimentally relevant length scales. For instance, we demonstrated that modeling solvent molecules implicitly by modulating the inter-molecular interactions in our forcefield, and implementing a very basic technique to simulate solvent evaporation leads to good experimental predictions at relatively low computational cost. Based on our observations, we therefore propose the following simulation guidelines for predicting the morphologies of OPV candidate molecules:Benchmark performance to identify the system size *N* that is practical for equilibrating hundreds of systems.Generate coarse phase diagrams with these inexpensive simulations to identify rough phase boundaries and interesting structures.Use simulated solvent evaporation to generate morphologies at experimental densities, with sufficiently large volumes.Validate predictions against experimental GIXS patterns, when available.

These guidelines can be applied to any OPV active layer material, and will help to ensure that the most information about model validity and OPV morphology are gained per unit of simulation time. Combining these guidelines with automatic identification methodologies and more detailed to more efficiently search parameter space [72,75] will further improve information gained per Central Processing Unit cycle. Extending the current investigation and applying these methods to a broader range of OPV candidate materials with potential for mitigating climate change will be the focus of future work.

## Figures and Tables

**Figure 1 polymers-10-01305-f001:**
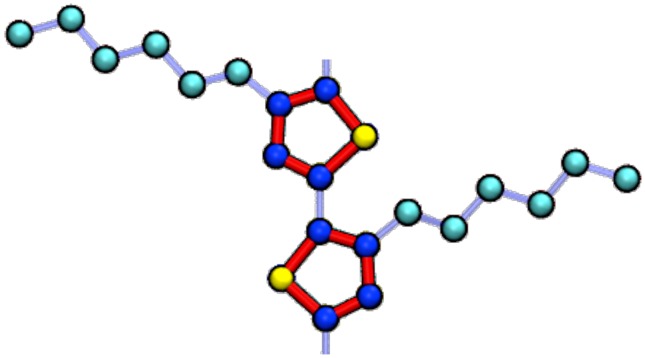
P3HT is modeled in this work with a united-atom representation. Sulfur beads (S) are yellow, aromatic carbon beads (CA) are dark blue, and aliphatic carbons beads (CT) are in cyan. Red bonds indicate thiophene rings modeled as rigid bodies, whereas the light blue indicate harmonic bonds.

**Figure 2 polymers-10-01305-f002:**
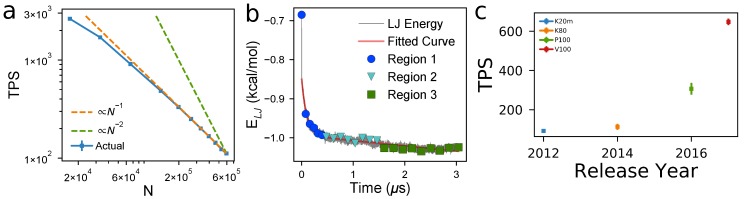
GPU-accelerated simulations of P3HT presented here achieve ideal performance scaling, but practical equilibration of a volume is limited by relaxation times. (**a**) computational performance measured by Timesteps Per Second (TPS, blue data) scales $O(N^{-1})$ (dashed orange line for simulations performed at T=600 K at ρ=0.72 g/cm^3^ and $\varepsilon_{s}=$ 0.8); (**b**) the time evolution of non-bonded potential energy shows a fast initial structural rearrangement (blue), a slower relaxation to equilibrium (cyan), followed by sampling of equilibrium microstates (green). The divergence of relaxation time (cyan) with system size, *N*, and density, ρ, puts practical limitations on the systems that can be equilibrated, despite high TPS values. (**c**) Computational performance, measured by TPS, has roughly doubled with each new hardware release for the last four generations of Nvidia hardware (K20m, K80, P100, and V100 cards). Error bars indicate one standard deviation over five independent simulations per cluster across multiple clusters.

**Figure 3 polymers-10-01305-f003:**

The degree of ordering, ψ, for Protocol (1) shows the most robust assembly occurs at lower densities, with more temperature-solvent combinations resulting in high ψ. (**a**) ρ=0.56 g/cm^3^; (**b**) ρ=0.72 g/cm^3^; (**c**) ρ=0.89 g/cm^3^; (**d**) ρ=1.05 g/cm^3^. Red regions denote order, whereas blue denotes disorder. Each black “x” indicates a measurement from an MD trajectory, and ψ values between measurements are linearly interpolated.

**Figure 4 polymers-10-01305-f004:**
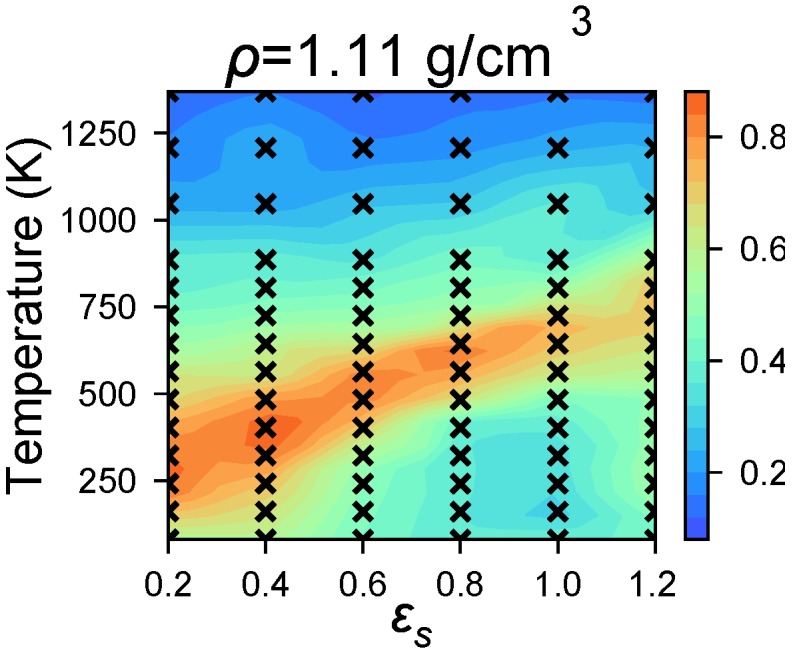
Morphologies sampled with Protocol (2) are observed to have higher ordering, ψ than Protocol (1) at the same state-points (compare Figure 3d), which suggests that simulating solvent evaporation helps to avoid long relaxation times.

**Figure 5 polymers-10-01305-f005:**
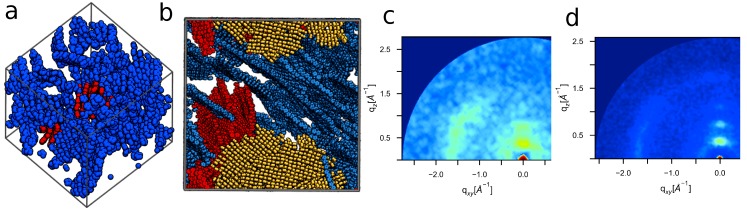
While small systems (**a**) are sufficient for identifying key structural features of ordered P3HT, large volumes (**b**) are needed to resolve structural periodicities ((**c**)-small, (**d**)-large) and therefore enable experimental validation.

**Figure 6 polymers-10-01305-f006:**
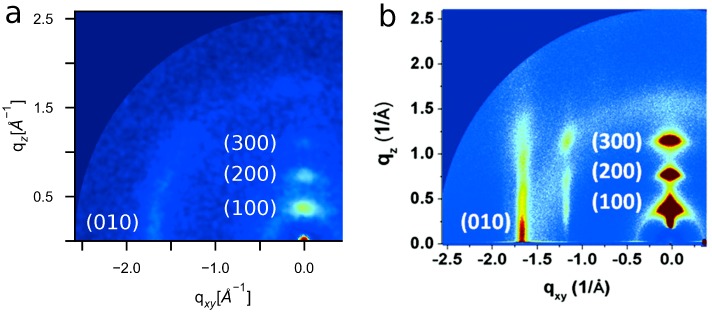
Our model produces (**a**) simulated GIXS patterns that closely match (**b**) experiment with π-stacking features along the (010) plane at 1.68 ± 0.02 Å^−1^ and alkyl-stacking features along the (100) plane with a spacing of 0.38 ± 0.02 Å^−1^. (Experimental GIXS pattern (**b**) reprinted with permission from Ref. [74]. Copyright 2012 American Chemical Society).

**Figure 7 polymers-10-01305-f007:**
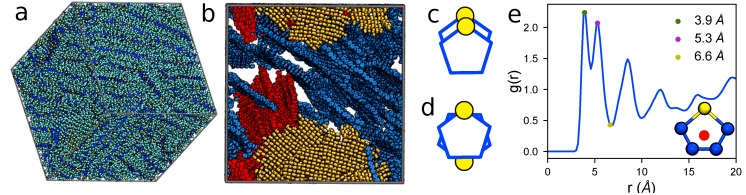
(**a**) a representative ordered molecular morphology of P3HT 15mers (CA-dark blue, S-yellow, CT-cyan) depicting π- and alkyl-stacked lamellae (state-point: implicit charges, T ∼ 600 K, $\varepsilon_{s}=0.8$, ρ=1.11 g/cm^3^); (**b**) the locations of the three largest crystallites in the system (colored blue, red, and yellow in order of descending size). Small crystallites and side chains are omitted for clarity. Within each crystallite, thiophene rings stack in (**c**) an aligned or (**d**) anti-aligned conformation, which are observed in (**e**) the RDF of the thiophene centroid (e-inset) as the green (3.9 Å) and magenta (5.3 Å) dots, respectively. The RDF minimum at 6.6 Å (yellow dot) is used as a clustering criterion describing the maximum separation of two rings in the same cluster.

**Table 1 polymers-10-01305-t001:** Optimized OPLS-UA interaction parameters for CA, CT, and S simulation elements used in this work. ϵ is the depth of the Lennard–Jones well, $\sigma_{LJ}$ is the van der Waals radius, and *m* is the mass.

Bead Type	$\sigma_{\mathrm{LJ}}$ (Å)	ϵ (kcal/mol)	*m* (amu)
CA	3.436	0.11	13.0
CT	3.905	0.17	15.0
S	3.436	0.32	32.0

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
