# Peer review of "Optimization and Validation of Efficient Models for Predicting Polythiophene Self-Assembly"

_polymers, 2018, doi:10.3390/polym10121305_

Reviewer 1 Report

The present manuscript describes and exemplifies a methodology to implement computationally-efficient and physically relevant molecular dynamics simulations of a self-assembly process, that in the specific case involves a polymer (P3HT) that is at the root of the current research focus on self-assembled organic photovoltaics. The work has a pronounced focus on computational efficiency, which leads the author to ponder rather interestingly on the many trade-offs required to balance physical significance with model simplification and handling (including eventual scalability). The results of the simulations for P3HT are then benchmarked with corresponding experimental observations obtained with Grazing Angle X-ray spectroscopy.

The manuscript is clearly structured and presented. The Introduction presents an interesting and rather broad context for the authors' work, including a comprehensive state-of-the-art and potential societal concerns addressed by organic photovoltaics (though we consider "climate-saving devices" (line 22) an hyperbole). The ensuing Model and Method sections are instructive and precise, as well as the exposition of Results in the fourth section, while the Conclusion section is currently a little weak. The quality of the writing is very high and commendable, the style concise and supported by significant illustrations (though they could be bigger in the final version). Finally, the manuscript is supported by an extensive technical section in the Supporting Information.

This paper addresses a very important topic that can certainly be of interest for a wide audience of the journal.
I recommend the work for publication after addressing some minor points

1) Please make sure that all acronyms be defined at first appearance in the text. For instance, Optimized Potential for Liquid Simulation is not (yet).

2) In Section 3.2 a paragraph is dedicated to the order parameter \psi, including technical references. However, from the text it is not clear how to appreciate a degree of order of, say, \psi = 0.4. The authors should possibly give an example, and/or add (in the main or supporting text) some of the more technical description of the references, to let the reader get a better appreciation of this and hence the color bars in Fig. 3 and 4.

3) The authors discuss the annealing protocols that they use to run their simulations, in particular with respect to volume shrinking and solvent evaporation. Here we feel a somewhat not completely resolved tension between a thermodynamic approach (reaching the equilibrium state for a given ensemble) and a transient analysis (relaxation dynamics, with consequences on volume scaling and simulation times). In particular, the authors admittedly report that their model for the solvent does not include entropic considerations; on the other hand, evaporation itself goes along with solvent flow, which in turn affects molecule motion and orientation. Provided the benchmark with experiments, it may seem that such effects can be to a first analysis negligible, but are they? We encourage the authors to comment more on such aspects along the lines here indicated, at the intersection between thermodynamics (entropic component of free energy, steric effects) and dynamics.

4) Also in view of the length of the manuscript, the authors should summarise in the first paragraph of the Conclusion sections their main results, both the general, MD-related ones and those specific to P3HT (which are present in the text at the end of the subsections). This will help the reader retain a clear message.

5) There are a few sporadic typos through the text (e.g. "into" is missing at the end of line 38; "The" to be cancelled in line 138). Also, the past instead of the present tense is recommended in the Conclusions.

Author Response

We thank both reviewers for taking the time to review our manuscript. In particular, we believe responding to Reviewer #1’s thoughtful feedback has helped strengthen this work. The reviewer’s suggestions are enumerated in the attached pdf, with our responses in red. We also upload two updated copies of our manuscript, with colorized.pdf showing the additions and removals, and paper.pdf with the uncolored updates incorporated.

Reviewer 2 Report

IThe paper can be accepted in present form.I believe this is a solid work that will have an impact in the field of OPV active layer material

Author Response

We thank both reviewers for taking the time to review our manuscript.